# Current Conceptualization and Operationalization of Adolescents’ Social Capital: A Systematic Review of Self-Reported Instruments

**DOI:** 10.3390/ijerph192315596

**Published:** 2022-11-24

**Authors:** Mikael G. Ahlborg, Maria Nyholm, Jens M. Nygren, Petra Svedberg

**Affiliations:** School of Health and Welfare, Halmstad University, SE-301 18 Halmstad, Sweden

**Keywords:** adolescents, instrument, measurement, psychometric properties, questionnaire, social capital, validation

## Abstract

There is a great heterogeneity in the conceptualization and operationalization of social capital in empirical research targeting adolescents. There has not yet been an attempt to systematically map and psychometrically evaluate the existing instruments for measuring social capital that have been developed and validated for adolescent samples. The aim of this systematic review was to identify and evaluate the design and psychometric properties of self-reported instruments for social capital, specifically developed and validated for use among adolescents. The design of this study was a systematic review guided by the COSMIN methodology for systematic reviews of Patient Reported Outcome Measures. The search included six electronic databases and no time frame was applied. Twenty studies were identified as describing the development and validation of a social capital instrument for adolescent samples. The results reveal common denominators, but also great variation in the design and validation of the instruments. Adolescents were only involved in the development procedures of four instruments. There is a lack of social capital instruments that cover both the multidimensionality of social capital and contextual relevance in relation to adolescents. Careful examination of instruments should thus precede a decision when designing studies and further instrument development involving the target group is encouraged.

## 1. Introduction

Social capital refers to the sum of resources that individuals access through their social networks [1]. A growing amount of research has explored the relationship between social capital and adolescent mental health since the beginning of the new millennium [2]. Social capital has in this period been shown to be considered as a valuable contributor when explaining inequalities in adolescent health, mental health, and health behaviours [3,4,5,6,7]. Based on findings such as these together with the persistent adverse developments in adolescent mental health [8], which are disproportionately affecting girls [9], research findings on the positive relationship between social capital and adolescent mental health outcomes thus need to be translated into policies and evidence-based interventions aimed at strengthening social capital. There are, however, major challenges to be faced in order to achieve this, which are linked to the conceptualization of social capital and its operationalization for measurement in research and practice. Research has shown that social capital has seldom been comprehensively conceptualized for adolescents as a group distinct from adults [2]. Social networks for adolescents differ from those of adults [10], and the pitfalls of not recognizing this include underestimating the agency of adolescents, overemphasizing parental influence, and the misinterpretation of how adolescents define important dimensions and factors that could be used for conceptualization [11]. A limitation in the literature is thus the lack of understanding of social capital concerning the target group [12] and how this understanding could be operationalized in a self-reported measurement. Standardized and validated measures of social capital can play a critical role in the work of promoting mental health in adolescents [13].

The association between social capital and adolescent mental health has been investigated using cross-sectional data, mostly with one or two indicators of social capital and various measurements for mental health outcomes [5,14,15,16]. Researchers and practitioners frequently choose items or sets of items that have not been developed to cover the multidimensionality of the concept and validated to assess self-reported social capital, but merely are linked to either social relationships, ties or networks [17]. Common sources of such items are, for example, the “Add Health Survey”, The National Longitudinal Survey of Youths and The Health Behaviour of School-aged Children Survey (where only the 2001/2002 survey explicitly focused on social capital) [18]. It is understandably tempting to apply the concept of social capital to large, national and international data sets, or to draw items from these well-acknowledged surveys. However, concern can be raised about a lack of transparency in the development and validation procedures as well as the inclusiveness of adolescents’ own voices during the process of development and psychometric validation, and in the perspectives representing the content of the end product. Social capital is a complex concept that also embodies multiple dimensions and constructs [19]. Bonding, bridging and linking social capital is generally used to distinguish between group contexts, reflecting social ties in homogenous groups, between heterogenous or cross-hierarchical groups [19]. Moreover, the structural dimension of social capital refers to the structure of networks, level of social participation and civic engagement, while cognitive social capital embodies how trust, reciprocity, sense of belonging and support are perceived by individuals [20]. Another distinction that is made frequently is a network or individual perspective, used to emphasize a collective force [20] or the resources that exist between individuals [1]. Using a single-dimension measurement to represent social capital may thus impair the usefulness of research findings. Similarly, researchers also encourage assessment of social capital in multiple relevant contexts such as family, school, peers, and neighbourhood/community when designing studies [21]. The school and peer context naturally overlap, since classmates constitute an important part of peer networks in adolescence. However, the school context is limited to physical location and time constraints and involves adults, while the peer context does not have those constraints [21]. By not considering the breadth of social capital, there is risk of missing out on important information that can contribute to a more comprehensive understanding of the relationship between social capital and adolescent mental health. It also raises the original question pertaining to psychometric validation: Are we measuring what we intend to measure?

The diversity in the conceptualization of social capital has led to a major debate within the field [22]. Some consider it to constitute a weakness, thus challenging the usefulness of the concept and the validity of the research findings. Others optimistically describe the diversity as vibrant [19], where various hypotheses are tested in order to provide a deeper understanding of the pathways between social capital and mental health [17]. What has become evident is that the lack of an agreed definition has led to great methodological heterogeneity in how social capital is operationalized for adolescents [2,5]. In summary, multi-dimensional and well-validated instruments for assessing social capital in adolescent samples should be more consequently used if the aim is to build an evidence base to be used to promote adolescent mental health. While the development and psychometric validation of new instruments for assessing social capital have been called for [17], it is first a necessity to investigate which instruments exist, what they measure and how they have been validated for adolescent samples. There has, to our knowledge, not been an attempt to systematically review and synthesize the evidence of self-report instruments for assessing social capital that have been developed and validated for use on adolescent samples. Similarly, little interest has been given to how adolescents have been involved in the development and validation processes of these instruments. To fill this gap, our current systematic review focuses on the evaluation of self-report instruments for assessing social capital among adolescents. We expect this systematic review to provide a direction for researchers, policymakers and practitioners on psychometrically validated measures of self-reported social capital in adolescent samples.

## 2. Materials and Methods

### 2.1. Design

The design of this study is a systematic review guided by the COSMIN methodology for systematic reviews of Patient Reported Outcome Measures (PROMs) [23]. This methodology provides a thorough description of how to evaluate the measurement properties once instruments are identified, as well as of the clearly defined steps from the initial search to the presentation. Steps 1–4 in the COSMIN manual concern the preparation and the performance of the literature search, as well as the selection of relevant studies, and Steps 5–7 concern the evaluation of the measurement properties of each instrument. 

### 2.2. Preparation and Performance of the Literature Search (Steps 1–4)

#### 2.2.1. Formulation of the Aim of the Review

A multidisciplinary research team within the disciplines of health science, public health and nursing and with expertise in social capital, mental health and youth studies was assembled to discuss the aim and identify research questions. The aim was formulated by clarifying the construct of interest (social capital), the population (adolescents), the type of instrument (self-report instruments developed and psychometrically validated for adolescent samples) and the measurement properties of interest. This process also included articulating the research questions. 

The aim of this systematic review was thus to identify and evaluate the design and psychometric properties of instruments for assessing social capital specifically developed and validated for self-reporting among adolescents (10–19 years). The specific research questions were: (1) What are the dimensions, constructs and contexts of interest within the instruments? (2) In which ways have adolescents been involved in the development and validation process of the instrument? and (3) How have the instruments been validated in terms of the face and content validity; internal structure; reliability and responsiveness?

#### 2.2.2. Criteria for Inclusion and Exclusion

The criteria for eligibility, which were thoroughly discussed between all the authors, were informed by the aim and the research questions. Studies eligible for inclusion were: instruments developed for or adapted to adolescent samples (10–19 years), explicit use of the term social capital in relation to instruments, description of the development and validation process, and explicit focus on adolescents as a group distinct from adults. Reasons for exclusion were: lack of included statement regarding item development or reference to the original source of items or instrument, lack of description of, or references to, a validation procedure that included adolescents, studies based on proxy reporting, such as by parents or others, and review articles. 

#### 2.2.3. Performance of the Search

Two librarians with expertise in search methodology were consulted on appropriate databases and search terms prior to performing the search. Based on this consultation and discussions between the authors, search words were tested in initial searches to assure accuracy and breadth. Keywords, titles and abstracts were then searched, accompanied by free text searches, in six electronic databases with a focus on the health and social sciences: PubMed, Scopus, Cumulated Index to Nursing and Allied Health Literature (CINAHL), PsycINFO, Sociological abstracts, and the Web of Science core collection. The final search words were: Adolescents OR Youth, Social Capital, Instrument (multiple synonyms combined with OR), development OR validat*. No time frame was applied, and articles published up until 8 February 2021 were included in the search. No restrictions were applied regarding the language of the publications, although an abstract available in English to enable initial inclusion was deemed necessary (see Appendix A for specifics). The systematic search was conducted in February 2021.

#### 2.2.4. Study Selection

All identified studies from the searches were imported to EndNote. Endnote X9 (Clarivate, London, UK, 2021) was used to facilitate managing of citations and identification of duplicates. A randomized sample of titles and abstracts were reviewed by all the authors to triangulate assessments and cement criteria in the first step of the study selection. Any uncertainties that arose during the process led to discussions between the authors and joint decisions on how to proceed. The first author (MA) then took the lead on conducting the screening process, reviewing all titles and abstracts. If any uncertainties arose during the screening process (Figure 1), a full-text version of the article was retrieved and the methods and results section briefly reviewed to ascertain details of the development and validation procedures, followed by a decision of inclusion to next step or exclusion. The information in some of the articles was either too imprecise to determine the eligibility or there was a statement that more information was available upon request from the authors. Both these issues were addressed by contacting the authors of these articles for any additional information or data that could strengthen the screening process during the selection phase. Any remaining issues on eligibility were discussed between at least two of the authors to assure consensus. The screening process resulted in a total of 54 articles that were subjected to a full-text review. Two authors (MA, MN) separately reviewed all the articles in full text, followed by a discussion between all the authors on whether each article was eligible for inclusion. This procedure resulted in the inclusion of 20 articles that met the eligibility criteria.

#### 2.2.5. Data Extraction

Three data extraction templates, inspired by those proposed in the COSMIN-methodology, were pre-developed to suit the aim and research questions and then used to systematically extract the data from each study. The first template was designed to extract study characteristics such as authors, publication year, journal of publication, country, aim and study design.

The second data extraction focused on answering the research question: what are the dimensions, constructs and contexts of interest within the instruments? Two strategies were applied to identify which dimensions and constructs of social capital that researchers had an explicit interest in measuring when developing their instrument, due to the diverse terminology used to describe social capital. First, any terms specified as dimensions or constructs were extracted from the studies. Second, these terms were interpreted into either a cognitive or structural dimension, as well as separated between bonding, bridging and linking forms of social capital [19]. The reason for this was to enhance the interpretability of the results. 

The third data extraction focused on answering the research question: In which ways have adolescents been involved in the development and validation process of the instrument? We were also interested here in understanding the setting and sample that were included in the development and validation process. The template thus included the sample (size, mean age and range and gender), setting, pilot sample, adolescents included in the development and/or in face validity. 

### 2.3. Evaluation of Measurement Properties of the Instruments (COSMIN Steps 5–7)

Two further data extraction templates were developed and used for the evaluation of measurement properties, inspired by the templates proposed in the COSMIN-methodology. The data extractions pertained to the research question: how have the instruments been validated in terms of face and content validity; internal structure; reliability and responsiveness? Each instrument was evaluated following Steps 5–7 in the COSMIN methodology, where Step 5 concerns content validity, Step 6 the internal structure and Step 7 involves reliability and responsiveness. All relevant information was systematically extracted from the articles into templates four and five following the aforementioned steps. The evaluation of content validation procedures, Step 5, resulted in the extraction of a number of subscales, their labels and total number of items accompanied by type of response options for the final versions of each instrument. Face and content validity evaluation identified whether the development of the instrument was theory explicit, if expert opinion was sought and if the target group was involved in the face validity procedures. The validation of internal structure of the instruments was evaluated in Step 6 involving structural validity, construct validity and concurrent validity or measurement invariance. The reliability and responsiveness in the studies were abstracted in Step 7. 

The validity concerning comparison with a golden standard was inapplicable to this evaluation, due to there not being any golden standard for the measurement of social capital. Abstaining from this procedure is an option presented in the COSMIN methodology [23] where inconsistent validation procedures and differing thresholds applied for analyses are described as acceptable reasons.

### 2.4. Citation and Further Validation of the Included Instruments

The article for each instrument was retrieved via Google Scholar as an additional step, and then, by using the function “cited by”, we explored whether the developed instruments had been included in any empirical studies where other validation procedures, in other contexts or age groups (still between 10–19 years old), had been conducted and reported. This step is not suggested in the COSMIN methodology [23], but this was carried out in order to determine whether any of the identified instruments had been tested for validity in other samples than in the original article, and if so, briefly present the findings. The Google Scholar search was undertaken on 1 June 2021.

## 3. Results

### 3.1. Study Characteristics

Twenty studies were identified that described the development and validation of instruments for assessing social capital in adolescent samples from an initial count of 1956 hits in the databases (see Table 1). The included instruments were validated in twelve different countries: the USA (n = 5), China (n = 2), Iran (n = 2), and one each from Australia, Brazil, Burundi, Colombia, Japan, Lebanon, the Netherlands, Spain and the Turkish Republic of Northern Cyprus. The year of publication ranged from 2005 to 2019. Twelve of the 20 instruments had an original design of items, four were adaptations of existing instruments, one was a mixture of adaptation and original item development and two included items from existing survey questionnaires. It was not made clear in one study [24] whether the items were drawn directly from existing survey questionnaires or if they had been developed by the authors and inspired by existing survey items. Multiple conceptualizations of social capital were found in the included studies; however, after reviewing references and theoretical underpinnings, most references were traced back to social capital as conceptualized either by Bourdieu [1], Coleman [25] or Putnam [20]. 

### 3.2. Dimensions, Constructs and Contexts of Interest

The dimensions, constructs and contexts of the instruments are described in Table 2. The cognitive dimension of social capital was represented in all the included instruments, mostly through the constructs of trust and sense of belonging, but also by social cohesion [24,26,29,33,38,40,41], reciprocity [28,34,35,43], support [24,30,35,44] and others. The structural dimension was represented by composition or structure of networks [29,33,36,42], interaction, participation or engagement [29,30,33,35,41,43] or similar. Furthermore, bonding social capital could be attributed to all instruments, while bridging social capital was represented in ten instruments [27,28,30,35,36,39,42] and linking social capital was only represented in one instrument [36] where the diversity of friendships could be interpreted as such. Regarding the contexts of interest in the instruments, the family (n = 9), school (n = 8), peer (n = 10) and neighborhood or community (n = 11) contexts were represented. The extended family or extended networks could be attributed to four instruments, while the online context was only found explicitly in one. No instrument represented social capital in all four occurring contexts, namely the family, school, peer and neighborhood or community. Incorporating social capital in at least two of these contexts was most common; however, single contexts were also presented [24,26,33,34,44].

### 3.3. Adolescents Involvement in the Development and Validation

There was a diversity of characteristics within some samples, but mainly between studies, since the included studies had differing aims and were conducted in different settings and countries. Rural and urban samples of adolescents from different ethnic and socioeconomic backgrounds were represented. Sample sizes ranged between 39 and 6853 with a median of 718 adolescents. The age ranges of the adolescents included in the studies varied but could be distinguished between early (10–13 years), mid (14–16 years) and late adolescence (17–19 years) (see Table 3). Five studies covered the entire span of adolescent ages in their samples [33,35,36,38,39]. The sample was limited to late adolescence in three studies [26,32,44]. Six studies had a sample covering mid-late adolescence [24,30,31,41,42,43] while only one study had a strict sample of mid adolescents [29]. The study by Hall [34] had a sample ranging from 6 to 16 years old, thus covering early to mid-adolescence. Lastly, four studies included only early adolescents in their validation study [27,28,37,40]. A large majority of the studies administered questionnaires for validation in a school setting. The exceptions distributed questionnaires in settings such as the household [24,35], at summer camps [36], both publicly and in school [39] and in two studies the setting was not specified [30,37]. Some studies described the setting and context further and specified, for example, art programme students [28], specific interest in sexual health [30], impoverished neighbourhoods [24], trauma exposure [34] and rural versus urban adolescents [29]. 

Adolescents were involved in the development phase of the instruments in only four studies. The involvement in three of them was described as target group interviews that were taken into consideration when items were decided upon [32,34,37]. The development of the questionnaire in the study by Onyx [39] was divided into two parts. Half of the items were drawn and adapted from an existing instrument and the other half was developed with help from adolescents. Involvement was described as active participation where a small group of adolescents had the leading role in developing questionnaire items. Focus groups interviews with the target group were conducted beforehand in the study by Harpham and colleagues [24], but there was no mention of linking this to the instrument development. Adolescent involvement for face validity purposes was more prevalent and described in 13 of the 20 studies. Face validity was mainly performed by having adolescents going through the questionnaire, pointing out any concerns followed by a discussion of the instrument in general. All face validity procedures were described as conducted with adolescents who were representative of the target group (Table 3).

### 3.4. Measurement Properties of the Instruments

#### 3.4.1. Content Validity

The instrument characteristics and content validity are presented in Table 4. The Cosmin Methodology describes that one of the key components of content validity is a clear conceptual description of the construct of interest that guides the development of the instrument. The evaluation of this aspect therefore relied upon description of, and references to, the theoretical underpinnings guiding development (here referred to as theory explicit). Four instruments met the expectations that the development of the instrument was theory explicit, expert opinion was sought and the target group was involved in face validity procedures [29,36,40,42]. Face and content validation procedures were absent in four instruments [24,26,27,35]. The remaining 12 instruments were validated using either one or two of these procedures (Table 4). Following face and content validation, most studies described how the phrasing and wording of the items were adjusted.

#### 3.4.2. Internal Structure

Seven instruments were not validated at all through structural validity analyses (see Table 5). Different forms of factor analyses were the primary method of choice among those which were. Principal component analysis (PCA) was used to establish and reduce dimensions for two instruments [26,29]. Exploratory factor analysis (EFA) was applied in four instruments with originally developed items and was followed by confirmatory factor analysis (CFA) [30,32,33,40]. Three instruments were structurally validated using only EFA [27,31,43], two by only CFA [38,42] and one by the use of hierarchical factor analysis [39]. Studies where factor analyses were performed primarily reported factor loadings, eigenvalues and total variance explained. Item factor loadings greater than 0.5 were reported to be desirable, but there were exceptions of lower thresholds, as low as 0.27 [30]. Factors with eigenvalues above 1.0 were consistent for instruments where values were reported. Variance explained was reported for seven instruments, with 32.8% [31] to 74.0% of variance explained by the cognitive subscale in the instrument [29]. Additional results included reporting model fit indices such as the comparative fit index (CFI) and root-mean-square error of approximation (RMSEA). Reported CFI values for factor analyses were all above 0.85. Values of RMSEA below 0.08 (indicating good fit) were reported for five instruments [30,32,33,38,42], while two instruments had values of 0.08 or above (mediocre fit) [39,40].

The evaluation of construct validity by comparing hypotheses of relationship to results confirmed positive or inverse relationships with dependent variables in line with hypotheses (*p*-values < 0.01) in seven studies (Table 5). Six studies reported inconsistent results or partly confirmed hypotheses of relationship with dependent variables. A test of normality (>5% in sample), which was interpreted as an indicator of construct validity, was used in one study to assess item content relevance, which resulted in retention of all 18 items [30]. A comparison of factors with those of a previous study from which the instrument was partly adapted was performed in another study [39]. Their findings indicated both recurring and unique factors. Factors derived from a factor analysis were compared to a theoretical framework in one study and consistency was found, although caution concerning the results was recommended due to a relatively low sample size [42]. Five instruments were validated for concurrent validity or measurement invariance with varying results (Table 5).

#### 3.4.3. Reliability and Responsiveness

Analyses of internal consistency and stability were used in the studies to assess reliability and responsiveness for the included social capital instruments. Cronbach’s alpha was the main indicator of internal consistency and was used in 18 of the 20 instruments. However, the way of reporting varied between the value of the total scale and each subscale respectively. Four studies reported the value for the total scale, of which only one had a value below 0.70 [35]. Among instruments with Cronbach’s alpha values reported for each subscale, the lowest values were found in the study by Almgren [26], ranging 0.28–0.86, and the highest in Takakura and colleagues’ [43] study, with values of 0.92 and 0.94 for the two subscales. Five instruments were tested for stability via test–retest procedures. The instrument developed by Carrillo-Álvarez and colleagues [29] showed an intra-class coefficient of 0.86, while Kappa-coefficients ranging 0.64–0.97 were reported in the study by Paiva et al. [40] for the four subscales. Two studies reported stability via correlation coefficient, ranging 0.48–0.81 for subscales in one study [24,43] and 67% of the 37 items above 0.70 in the other [23]. Krasny and colleagues [36] compared mean scores with a control group and found no significant differences (see Table 5).

#### 3.4.4. Citation and Further Validation of the Included Instruments

In total, the 20 included studies were cited 647 times (range 1–90) in Google Scholar. The articles were scanned for additional validation procedures, but only two of the twenty instruments had been used in other studies that included validation procedures. First, the instrument developed by Onyx and colleagues [39] was translated and validated in a sample of mid to late adolescents in Greece in the study by Koutra et al. [45]. Factor analysis produced a five-factor solution as opposed to the seven-factor solution in the original study, and four items were dropped from the original 34 items. The variance explained for the instrument was about 10 percentage points lower than in the original study and Cronbach’s alpha ranged 0.53–0.73 for subscales. The other instrument was the one developed by Paiva and colleagues [40], which was used in three studies, all with the aim of translating and validating the instrument. These studies involved one Chinese version [46], validated for 10–12-year-olds; one Japanese version [47], validated for 10–15-year-olds; and one Persian version [48] validated for mid to late adolescents. More specifics can be found via references.

## 4. Discussion

This systematic review identified 20 instruments for assessing social capital that were developed and validated for adolescent samples. The thorough examination and evaluation of these instruments revealed some conceptual drawbacks and incoherence. First, about half of the studies did not provide explicit theoretical underpinnings described to guide the development of the instrument, the dimensions and constructs of interest and item generation. This can only be considered a vital flaw, since researchers within this field are encouraged to specify how they envisage the concept of social capital and build on existing theories to make findings more conceptually sound and empirically useful [12]. Second, the use of the term social cohesion to label constructs of interest and subscales was somewhat surprising given that social cohesion is often described as a broader concept, in which social capital acts as an important ingredient [19,49]. Lastly, some instruments included constructs or subscale labels that are not traditionally associated with social capital, but instead stem from theories on individual-level psychology, e.g., self-concept and self-efficacy. This raises concern about conceptual coherence in an already heterogenous environment of self-reported measures of social capital.

The common denominators for the included instruments were the inclusion of constructs representing the cognitive dimension, as well as bonding social capital. It is reasonable to suggest that trust, sense of belonging, reciprocity and support represent core constructs that should be included in some way when assessing social capital among adolescents based on our findings. There was, however, a great variation in the design of the questionnaires, number of items and subscales and validation procedures. Although a grey zone was encountered in the distinction between what counts as original item development and drawing on or adapting items from existing measures, twelve described procedures were interpreted as inclusive of original item development. This distinction was facilitated by information on whether data gathering was prospective in relation to the development or if items were drawn from retrospective data. Moreover, conducting a pilot study and including adolescents in face validity procedures also indicated original item development. Five instruments were adaptations of existing instruments for investigating social capital, all of which had previously been validated for adult samples. Of these, one instrument [27] did not undergo face validation procedures before the study was performed. If this truly is the case, it can be considered a weakness. 

The instruments, with only one exception, targeted social capital in either one or several of the contexts of family, school, peers and neighbourhood/community. The exception [44], focussed on the online context, but since peer support was also the focus of that study, it is possible to attribute the peer context to this instrument as well. No instrument assessed social capital in all four contexts and only one instrument covered both the cognitive and structural dimensions together with bonding, bridging and linking social capital. This supports the conclusions that other researchers have made that it is unlikely for a single instrument to be able to cover the multidimensionality of the concept in adolescent samples [21]. However, the almost non-existence of linking social capital in these instruments may be an indication of its possible irrelevance for adolescents as a group and modest immediate effects on health outcomes, which at least may be true for early and mid-adolescence. Qualitative research has shown that older adolescents lack linking forms of social capital, which in the absence of strong social capital in the family, would help them navigate through structural systems and help them in their transition to adulthood [50].

The involvement of adolescents in the developmental phase of the instruments was infrequent and only seen in four studies. Only one of these incorporated adolescents as active agents in the item construction. This fact highlights the confidence that researchers have in previous research and a dominating view of theory as the main source of knowledge for operationalizing social capital. In order to stay true to what has historically been described as social capital, this approach may perhaps be motivated. However, the social environment and the ways in which adolescents socialize and communicate have undoubtedly changed over the past decades alongside societal advancements in mobility, communications and the economy. Arguably, so has also the relationship with adolescent health and wellbeing [51]. To maintain a possibility of discovering sociocultural specific indicators of social capital or new universal resources, we thus propose at least interviewing adolescents of the target group as part of the development procedure of new instruments. Due to the emphasis on consideration of sociocultural context when measuring social capital [12], involvement of adolescents in face validity procedures provides an important contribution in terms of validation. There is thus cause to question the validity of the instruments where such procedures were not reported. Nonetheless, some of these instruments comprised items drawn from existing surveys, where larger questionnaires covering multiple concepts may have been previously validated. Experience from this systematic review provides the insight that such procedures are more difficult to perform than in development and validation studies of strict instruments for assessing social capital with originally developed items. 

A review of the results of all the evaluation steps revealed that the most comprehensive and transparent validation procedures were found for the Family Social Capital Questionnaire [29] and the Social Capital Questionnaire for Adolescent Students [40]. These provided content validity through explicit theory, expert opinion and target group opinion, and analyses for structural validity, internal consistency and stability. Most instruments adequately established face validity, structural and construct validity and internal consistency as described in the COSMIN methodology [23] although with varying levels of transparency. The instruments evaluated in this systematic review covered differing and multiple dimensions of social capital in varying contexts and settings. Not one instrument could be described as flawless, and all instruments were indeed validated for adolescent samples. We thus recommend that researchers and practitioners consider either of these when designing new studies where social capital is to be measured in a sample of adolescents. However, since many of the evaluated instruments were developed within a specific setting and with the intention of investigating the relationship between social capital and a certain outcome, this should be of guidance for such discussions. With reference to specific outcomes, the initial rationale for this systematic review was a search for validated instruments of social capital in order to investigate the relationship between social capital and adolescent mental health. In a hypothesized scenario, choosing an appropriate instrument would involve careful consideration of context, age, measurement properties and validation procedures, but also the original intention underlying instrument development. Six instruments were developed to assess social capital in its relationship with mental health outcomes [24,33,34,37,38,43]. These instruments were validated for different age groups through diverse procedures and would therefore need close examination before determining appropriateness. However, the two instruments with the most comprehensive procedures [29,40] were developed to assess social capital in general and, though with their own limitations, could therefore be regarded as universal in relation to outcomes. 

If more researchers and practitioners are made aware of existing validated instruments, the usage of these may increase, which would enhance the comparability and transferability of findings. Even though the instruments dated back to 2005, other studies where any of these instruments had been used and validation procedures had been provided of their own were rare, as revealed by the citation investigation. This review also reveals the need for validated instruments that capture the multidimensionality of the concept, covering all four contexts of the family, school, peers and community. The difficulties of translating research findings into evidence-based policy and effective interventions will likely persist until these have been produced. Given the efforts of reducing mental health problems and promoting adolescent mental health worldwide, strengthening adolescent social capital should be a priority.

### Limitations

Not all the steps described in the COSMIN methodology could be performed in this systematic review. The patient-reported outcome measures more often adhere to areas where a golden standard is present and where the outcome measure is more stringently defined, such as in the field of medicine. Nevertheless, we found the methodology to provide a robust framework that was fitting for the aim of our study. As mentioned, there are quite a few studies in which social capital has been assessed in adolescent samples that were not included in this review. Some of them provided an internal consistency analysis for their measure but were not eligible simply because of this. There is, however, a possibility that by adding databases covering other research fields, additional instruments could have been identified. We limited this risk by consulting experts on search strategies. Six of the included instruments were validated in samples that involved participants below the age of 10 or exceeded 19 years. Although individuals outside this range were a minority in each sample, it is necessary to mention since this may affect the results of the validation procedures. 

All the authors were involved in the procedure of each step and any uncertainties that arose were discussed until consensus was reached. By having all authors initially review a randomized sample of studies independently followed by discussion, the reliability of the screening procedure was increased. While one author took the lead in performing the screening and evaluation, discussions preceded each step and were held along the way, and any issues were resolved amongst all authors. To further enhance the trustworthiness of this study, we contacted a number of authors in cases where issues were not resolved amongst the authors of this study and received multiple responses, for which we are grateful.

## 5. Conclusions

This systematic review identified 20 instruments for assessing social capital that were developed and validated for adolescent samples. The common denominators for the included instruments were the inclusion of constructs representing the cognitive dimension, as well as bonding social capital. Apart from these, there was great variation in the design of the questionnaires, number of items and subscales and validation procedures. There was no instrument that assessed social capital in the contexts of the family, school, peers and neighborhood or community. Adolescents were only involved in the development of four studies, which testifies to a predominantly theory-driven approach in the design of instruments for assessing social capital. Nevertheless, the theoretical underpinnings guiding the development procedures were poorly described for many of the instruments. Two instruments stood out in terms of their transparency and the adequate reporting of the validation procedures, both of which aimed to assess social capital in general and not in relation to a specific outcome.

Six instruments were developed to assess social capital in the relationship with mental health outcomes; however, these instruments were validated for different age groups through diverse procedures and would therefore need closer examination before determining their appropriateness in research and practice. Further validation work with the target group is proposed and the development and validation of new social capital instruments for adolescent samples is encouraged.

## Figures and Tables

**Figure 1 ijerph-19-15596-f001:**
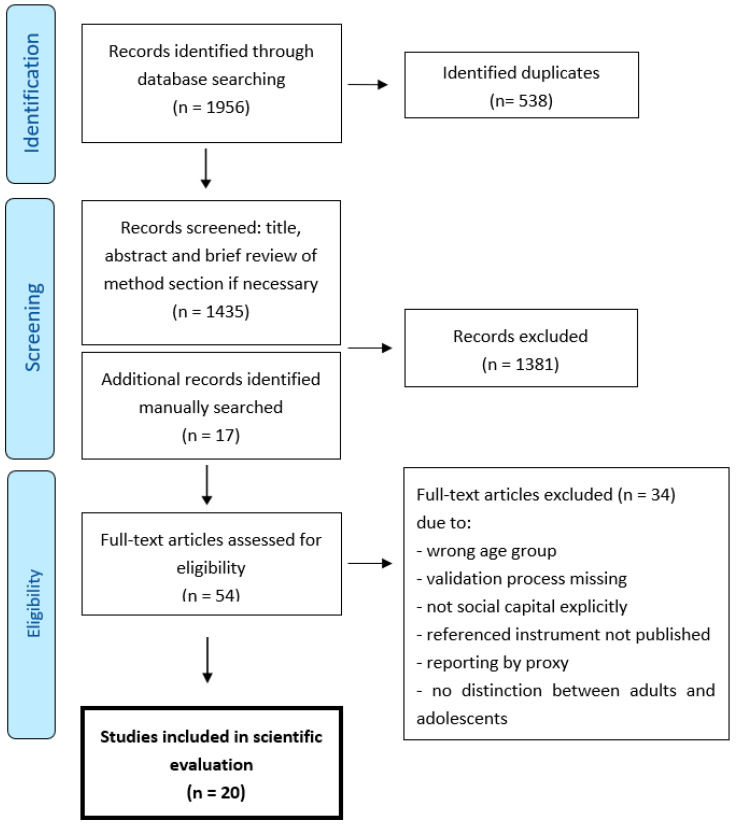
Flowchart diagram of study selection process.

**Table 1 ijerph-19-15596-t001:** Study characteristics.

Authors	Year	Journal of Publication	Country	Aim/Research Question	Study Design
Almgren, Magarati & Mogford [26]	2009	Journal of Adolescence	USA	Whether there is an explanatory contribution of social capital to the self-reported health of adolescents that adds to the variance explained by demographic and developmental covariates.	Quantitative, cross-sectional
Antheunis, Schouten & Krahmer [27]	2016	Journal of Early Adolescence	Netherlands	To examine the role of social networking sites (SNSs) in early adolescents’ social lives	Quantitative, cross-sectional
Buys & Miller [28]	2009	International Journal of Education & Arts	Australia	To better understand how and if participating in CCD initiatives lead by an independent youth arts organization impacts the development of social capital in school children residing in a socio-economically disadvantaged area of South-East Queensland, Australia	Mixed-methods, cross-sectional
Carrillo-Álvarez, Villalonga-Olives, Riera-Romaní & Kawachi [29]	2019	SSM-Population Health	Spain	To develop a Questionnaire on Family Social Capital (FSCQ) for use in an adolescent population and to test its reliability and validity.	Mixed-methods, cross-sectional
Cordova, Coleman-Minahan, Bull & Borrayo [30]	2019	Youth & Society	USA	To develop and examine the factor structure of the Brief Social Capital for Youth Sexual and Reproductive Health Scale	Quantitative, cross-sectional
Curran [31]	2007	Journal of Alcohol and drug Education	USA	To examine the relationship between social capital and substance use by high school students	Mixed-methods, cross-sectional
Ergün, Uzunboylu & Altinay [32]	2018	Quality & Quantity	Turkish Republic of Northern Cyprus	To investigate the connection between school climate and students’ social capital development	Mixed-methods, cross-sectional
Geraee, Eslami & Soltani [33]	2019	Health Promotion Perspectives	Iran	To investigate the direct and indirect relationships between family social capital and life satisfaction, and the possible mediating role of social media use between the variables among Iranian adolescents	Mixed-methods, cross-sectional
Hall, Tol, Jordans, Bass & de Jong [34]	2014	Social Science & Medicine	Burundi	To examine the longitudinal association between cognitive social capital and mental health (depression and posttraumatic stress disorder (PTSD) symptoms), functioning, and received social support of children in Burundi	Mixed-methods, longitudinal
Harpham, Snoxell, Grant & Rodriguez [24]	2005	British Journal of Psychiatry	Colombia	To measure the prevalence of common mental disorders among low-income young people in the city of Cali, Colombia and to examine associations with violence and social capital	Quasi-experimental, cross-sectional
Khawaja, Abdulrahim, Soweid & Karam [35]	2006	Social Science & Medicine	Lebanon	To examine the association between place and components of social capital among adolescents living in three impoverished communities outside of Beirut, the capital city of Lebanon	Quantitative, cross-sectional
Krasny, Kalbacker, Stedman & Russ [36]	2013	Environmental Education Research	USA	To develop and test for reliability a survey to measure cognitive and structural attributes of social capital among youth	Quasi-experimental, cross-sectional
Lau & Li [37]	2011	Children and Youth Services Review	China	To examine the extent to which variations in family and school social capital can be explained by child’s differing socioeconomic and demographic background and school characteristics; and second, the extent to which family and school social capital in combination might be associated with variations in child subjective well-being in Shenzhen, China	Mixed methods, cross-sectional
Magson, Craven & Bodkin-Andrews [38]	2014	Australian Journal of Educational & Development Psychology	Australia	To (1) develop a new multidimensional measure of social capital that accurately quantifies the extent of bonding, bridging, and linking capital an individual possesses; (2) test the psychometric properties of the new measure based on confirmatory factors analyses, tests of reliability, and invariance, and (3) establish the convergent validity of the new measure by examining the associations between the Social Capital and Cohesion Scale factors and mental health constructs	Quantitative, cross-sectional
Onyx, Wood, Bullen & Osburn [39]	2005	Youth Studies Australia	Australia	To report on a project in which young people were actively involved in identifying relevant items for a social capital scale, administering a questionnaire concerning social capital and other social issues, and collating the results	Mixed-methods, cross-sectional
Paiva, de Paiva, de Oliveira Filho, Lamounier, Ferreira, Ferreira, et al. [40]	2014	PLoS One	Brazil	To develop and validate a quick, simple assessment tool to measure social capital among adolescent students.	Mixed-methods, cross-sectional
Pourramazani, Sharifi & Iranpour [41]	2019	Addict Health	Iran	To determine the prevalence and the relationship between SC and substance use in Southeast Iranian adolescents	Mixed methods, cross-sectional
Ryan & Junker [42]	2019	Youth & Society	USA	To measure the multidimensional concept of social capital among youth in the domain of postsecondary transitions	Mixed-methods, cross-sectional
Takakura, Hamabata, Ueji & Kurihara [43]	2014	School Health	Japan	To develop self-rating scales of social capital at school and neighborhood among young people and to evaluate psychometric properties of the scales.	Quantitative, cross-sectional
Wang & Gu [44]	2019	Asia Pacific Journal of Education	China	1. What is the status of social media use of Chinese adolescents in terms of frequency, place, type and aim? 2. Whether and to what degree does online social capital influence academic identity? 3. How do demographic variables influence the relationship between online social capital and academic identity?	Quantitative, cross-sectional,

**Table 2 ijerph-19-15596-t002:** Type of instrument, conceptualization, dimensions, constructs and contexts.

Reference	Type	Conceptualization	Dimensions/Constructs of Interest	Contexts Specified	Cognitive	Structural	Bonding	Bridging	Linking
Almgren et al., (2009) [26], USA	Items drawn from existing survey	Inspired by Baum & Ziersch (2003), Wilkinson (2009).	Local opportunity structure and social cohesion	School	x		x	x	
Antheunis et al., (2016) [27], The Netherlands	Adaptation of the Internet social capital scales, Williams (2006)	Inspired by Bourdieu & Wacquant (1992), Putnam (2000)	Bonding and Bridging social capital	Peer, Community	x	x	x	x	
Buys & Miller. (2009) [28], Australia	Adaptation of the social capital questionnaire, Onyx & Bullen (2000)	Onyx & Bullen (2000), not further specified	Self-concept, reciprocity; extended networks; feelings of obligation; feelings of trust and safety.	peer, school, extended network	x	x	x	x	
Carrillo-Álvarez et al., (2019) [29], Spain	Original item development	Inspired mainly by Coleman (1988), Litwin (2014) and Widmer et al., (2013)	Structural: Network structure; Quality of ties; Social interaction. Cognitive: family cohesion; sense of belonging; informal control or collective efficacy	Family, extended family	x	x	x		
Cordova et al., (2019) [30], USA	Original Item development	Inspired by Furstenberg & Hughes (1995); Lochner et al., (1999).	Civic engagement; adult support; community support	Community, peer	x	x	x	x	
Curran (2007) [31], USA	Items drawn from existing survey	Inspired by Lin (2001) and Kreutzer & Lezin (2002).	Not specified	Family, family-school connection	x	x	x		
Ergün et al., (2018) [32], Turkish republic of Northern Cyprus	Original item development	Inspired by qualitative interviews and literature review.	Trust, respect and affection	Peer, family	x		x		
Geraee et al., (2019) [33], Iran	Original item development	Inspired by literature review and expert interviews.	Family functioning, family composition, family cohesion, family interactions	Family	x	x	x		
Hall et al., (2014) [34], Burundi	Original item development	Inspired by De Silva et al., (2006), Bourdieu (1986), Inaba (2013) and formative qualitative work	Cognitive social capital, trust, cohesion and reciprocity	Community	x		x		
Harpham et al., (2005) [24], Colombia	Unclear	inspired by SCAT (World Bank, Krishna & Schrader, 1999) and the World values survey	Trust, social cohesion, support and reciprocity, social control, civic participation	Community	x	x	x	x	
Khawaja et al., (2006) [35], Lebanon	Original item development	Inspired by Putnam (1993, 2000) Literature review to establish dimensions and extract items	Civic engagement and community involvement; locational capital; trust; reciprocity; social support; and social network	Community, peer, family, extended family	x	x	x	x	
Krasny et al., (2013) [36], USA	Adaptation of the National Social Capital Benchmark study.	Inspired mainly by Putnam (1995)	Trust; informal socializing; diversity of friendships; associational involvement; civic leadership	Community, peer, school	x	x	x	x	x
Lau & Li (2011) [37], China	Original item development	Inspired by focus group interviews with parents of target group. Multiple theorists referenced	Structural and cognitive social capital	Family, school	x	x	x		
Magson et al., (2014) [38], Australia	Existing and original item development	Inspired by Stone (2001) and Stone & Hughes (2002), Putnam (2000)	Trust, sense of belonging, social cohesion	peer, family, community	x		x	x	
Onyx et al., (2005) [39], Australia	Partly adaptation of the social capital questionnaire Onyx & Bullen (2000)/Original item development	Inspired by literature review and target group involvement	Not specified	Peer, community,	x	x	x	x	
Paiva et al., (2014) [40], Brazil	Original item development	Inspired by Coleman (1988) and literature review	Social cohesion and Trust	Peer, school, community	x		x		
Pourramazani et al., (2019) [41], Iran	Original item development	Inspired by Grootaert et al., (2004) Harpham et al., (2005) Paiva et al., (2014)	Trust, social participation, social cohesion, bonding SC	School, community, family	x	x	x		
Ryan & Junker (2019) [42], USA	Original item development	Inspired by Lin (2001)	Network structure: closeness, trust, network density, network norms, belongingness. Network content: access to resources.	Family, peer, extended network, school	x	x	x	x	
Takakura et al., (2014) [43], Japan	Original Item development	Definition from Inaba (2013). inspired by Morgan & Haglund (2009), Boyce et al. 2008, Elgar et al., (2010) among others	Cognitive: trust and reciprocity, Structural: social participation	School, neighborhood	x	x	x		
Wang & Gu (2019) [44], China	Adaptation of the Internet social capital scales, Williams (2006)	Not specified	Bonding social capital: Emotional and substantive support	Online	x		x		

**Table 3 ijerph-19-15596-t003:** Sample, setting and involvement of adolescents in development and validation.

Reference	Sample			Setting	Pilot Sample	Adolescents Included in Development	Adolescents Included in Face Validity
	N	Mean Age (Range)	Gender % Female				
Almgren et al., (2009) [26], USA	6853	(17–18)	55.0%	Administered in school, adolescents in general, group differences	N/A *	No	No
Antheunis et al., (2016) [27], Netherlands	3068	13.46 (11–14)	53.7%	Administered in school, adolescents in general	N/A	No	No
Buys & Miller. (2009) [28], Australia	39	10.6 (9–13)	69.2%	Administered in school, art programme students, intervention evaluation	12 students, target group	No	Yes, target group
Carrillo-Álvarez et al., (2019) [29], Spain	429 (245 + 184) (59 retest)	(14–16)	54.3%	Administered in school, adolescents in general, rural vs. urban comparison	See sample	No	Yes, target group
Cordova et al., (2019) [30], USA	200	17.4, (14–21)	57.2%	Not specified, Sexual health, Impoverished neighborhoods, ethnic minority residents	See sample	No	No
Curran (2007) [31], USA	590	(14–18)	50.2%	Administered in school, risk and protective factors in adolescents	N/A	No	Yes, target group not specifically for SC instrument
Ergün et al., (2018) [32], Turkish republic of northern Cyprus	304	(17–18)	Not specified	Administered in school, adolescents in general	40 students, target group	Yes, 15 target group interviews	Yes, target group
Geraee et al., (2019) [33], Iran	835	15.2 (12–19)	48.7%	Administered in school, adolescents in general	See sample	No	Yes, target group
Hall et al., (2014) [34], Burundi	176	12.0 (6–16)	46.6%	Administered in school, adolescents exposed to PTEs with mental health problems	See sample	Yes, target group interviews	Yes, target group
Harpham et al., (2005) [24], Colombia	1057	(15–25) (70.9% 15–20)	57.30%	Household setting, Impoverished neighborhoods, mental disorders	N/A	No	No
Khawaja et al., (2006) [35], Lebanon	1294	(13–19)	Not specified	Household setting, Impoverished neighborhoods	N/A	No	No
Krasny et al., (2013) [36], USA	210 + 87	(10–18)	48–57%	Administered at summer work camp, Social capital in relation to Environmental Education	9 adolescents, 14–18 years	No	Yes, target group
Lau & Li (2011) [37], China	1306	(11–12)	43.9	Not reported, early adolescents in general	Two rounds, not specified	Yes, target group interviews	Yes, target group
Magson et al., (2014) [38], Australia	1371	(12–17)	38.7%	Administered in school, adolescents in general, relation to mental health	N/A	No	No
Onyx et al., (2005) [39], Australia	173	(12–20)	47.9%	Administered publicly and in school, rural adolescents	See sample	Yes, target group, active participation	Yes, target group
Paiva et al., (2014) [40], Brazil	101	12 (12)	53.5%	Administered in school, adolescents in general	12 students, target group	No	Yes, target group
Pourramazani et al. [41], (2019), Iran	600	16.63 (15–18)	54.8%	Administered in school, relation to substance use	28 students, target group	No	Yes, target group
Ryan & Junker (2019) [42], USA	140	(14–18)	61%	Administered online in school setting, post-secondary transition	See sample	No	Yes, target group
Takakura et al., (2014) [43], Japan	1241	(15–18)	55%	Administered in school, adolescents in general	N/A	No	No
Wang & Gu (2019) [44], China	1286	18.9 (18–20)	60.3%	Administered online, Retrospective questions, online SC	78 students,	No	Yes, target group

* N/A = Not Applicable.

**Table 4 ijerph-19-15596-t004:** Instrument characteristics and face and content validity (COSMIN step 5).

					Face and Content Validity
Reference	Instrument Name	Subscales/Number of Items	Subscale Labels	Response Options	Theory Explicit/Expert Opinion/Target Group Opinion/Revision
Almgren et al., (2009) [26]	Not named	4/13	Positive school affiliation, safe learning environment, social network cohesion, parents having knowledge of friends’ plans	Likert scale, not further specified	Not reported
Antheunis et al., (2016) [27]	Not named	2/7	Bridging and Bonding social capital	5-point Likert scale	Not reported
Buys & Miller (2009) [28]	Not named	4/22	Self-concept, reciprocity; extended networks; feelings of obligation; feelings of trust and safety.	Dichotomous	Target group opinion, revision
Carrillo-Álvarez et al., (2019) [29]	Family Social Capital Questionnaire (FSCQ)	2/24 + 7	Structural: Structure of the network; Quality of the ties; Social interaction. Cognitive: Collective efficacy; Informal control; Sense of belonging; Family conflict (Bridging SC as supplement)	Multiple choice, 6-point Likert scale	Theory explicit, expert opinion, target group opinion, revision
Cordova et al., (2019) [30]	Brief Social Capital for Youth Sexual and Reproductive Health Scale (BSC-Youth)	3/16	Community support and condom self-efficacy; Adult support; Civic engagement	5-point Likert scale	Theory explicit
Curran (2007) [31]	Not named	4/39	Parental rules and expectations; human capital; family climate; family connectedness	Multiple choice	Content validity assessed for the complete YRPS, not social capital
Ergün et al., (2018) [32]	Social capital scale	4/22	Trust in friendships; Interaction in the family; Sensitivity in friendships; common social capital scale	5-point Likert scale	Expert opinion, target group opinion, revision
Geraee et al., (2019) [33]	Family social capital scale	4/31	Family cohesion; family interactions: lack of family conflicts: family control	5-point Likert scale	Expert opinion, target group opinion, revision
Hall et al., (2014) [34]	Not named	6 items	Cognitive social capital	4-point Likert scale	Theory explicit, target group opinion/revision
Harpham et al., (2005) [24]	Not named	6/37	Trust in institutions; trust in people; social cohesion; solidarity; social control; civic participation	3- and 5-point Likert scale	Not reported
Khawaja et al., (2006) [35]	Not named	6/18	Civic engagement and community involvement; locational capital; interpersonal trust; reciprocity; hypothetical social support; and social network	4 and 5-point Likert scale, Dichotomous, multiple choice,	Not reported
Krasny et al., (2013) [36]	Not named	5/27	Social trust; informal socializing; diversity of friendships; associational involvement; civic leadership	5-point Likert scale, Dichotomous	Theory explicit, expert opinion, target group opinion, revision
Lau & Li. (2011) [37]	Not named	6/38	Bond between children and parents (structural and cognitive), teacher-student relationship, peer relationship (structural and cognitive), bonds between parents and schools	4 and 5-point Likert scale	Target group opinion, revision
Magson et al., (2014) [38]	Social Capital and Cohesion Scale (SCCS)	6/29	Family SC, Peer SC, Neighbor SC, institution SC, Belonging, Isolation	5-point Likert scale	Theory explicit
Onyx et al., (2005) [39]	Youth Social Capital Scale	7/34	Factors labelled connections with friends; participation in the community; moral principles; neighborhood connections; trust and safety; belonging with a group of friends; youth social agency	4-point Likert scale	Target group developed/opinion, revision
Paiva et al., (2014) [40]	Social Capital Questionnaire for Adolescent Students (SCQ-AS)	4/12	School Social Cohesion; School Friendships; Neighborhood Social Cohesion; Trust: school/neighborhood	3-point Likert scale	Theory explicit, expert opinion, target group opinion, revision
Pourramazani et al., (2019) [41]	Not named	6/36	Social trust; social participation; social cohesion; bonding with neighbors; Bonding with family; Bonding with schools	5-point Likert scale	Expert opinion, target group opinion, revision
Ryan & Junker (2019) [42]	Not named	4 + 1/17 + 4	Network structure: network location; collective assets peer norms; sense of belonging. Network content: access to resources.	Name generator, Dichotomous, 4 and 5-point Likert scale	Theory explicit, expert opinion, target group opinion, revision
Takakura et al., (2014) [43]	Not named	2 subscales/Cognitive 12 items, Structural not specified	Cognitive social capital at school/neighborhood; Structural social capital at school/neighborhood	5-point Likert scale, 6-point scale	Theory explicit
Wang & Gu (2019) [44]	Not named	2/6	Online social capital: Emotional and substantive support	5-point Likert scale	Target group opinion/revision

**Table 5 ijerph-19-15596-t005:** Internal structure, reliability and responsiveness of instruments (COSMIN steps 6–7).

	Internal Structure						Reliability and Responsiveness			
Reference	Structural Validity		Construct Validity/Hypotheses Testing/Convergent Validity		Concurrent Validity/Measurement Invariance		Internal Consistency		Stability	
	Factor Analysis	Results	In Line with Hypothesis/Subgroup Comparison	Results	CFA or DIF Analyses	Results	Cronbach’s Alpha/KR	Results	Test-retest/ICC/Kappa/Weighted Kappa	Results
Almgren et al., (2009) [26]	PCA	Four-factor solution, eigenvalues >1, item loadings >0.4.	Positive relationship with self-rated health	inconsistent	N/A		Cronbach’s alpha	Subscales range 0.28–0.86	N/A	
Antheunis et al., (2016) [27]	EFA	Two-factor solution, eigenvalues 2.36 for bonding and 1.94 bridging, variance explained 33%.	Positive relationship between social capital and SNS use intensity	*p* < 0.001	N/A		Cronbach’s alpha	Bonding 0.76, bridging 0.66	N/A	
Buys & Miller (2009) [28]	N/A		Positive relationship with art program	inconsistent	N/A		N/A		N/A	
Carrillo-Álvarez et al., (2019) [29]	PCA, then CFA in different sample	Seven-factor solution (loadings > 0.5) eigenvalues range 1.205–4.045, variance explained 64.8% and 74.0%. CFI 0.94.	See concurrent validity		Mean score comparison between rural and urban group	Significant difference for structural dimension *p* < 0.01	Cronbach’s alpha	Structural 0.79, cognitive 0.79	Test–retest with ICC	0.86
Cordova et al., (2019) [30]	EFA and CFA	Three-factor solution, CFA factor loadings 0.27–1.15, CFI 0.90, RMSEA 0.068	Tests of normality	all items >5% occurrence in sample	N/A		Cronbach’s alpha	Community support and condom S-E 0.60, adult support 0.83, civic engagement 0.59	N/A	
Curran (2007) [31]	EFA	Four-factor solution, Eigenvalues > 1.0, variance explained 32.8%	Inverse relationship with substance use	*p* < 0.001	N/A		Cronbach’s alpha	Subscales range 0.41–0.80	N/A	
Ergün et al., (2018) [32]	EFA, VFA and CFA	EFA: variance explained 51.50%, CFA: factor loadings > 0.05, RMSEA 0.05, CFI 0.91	Correlation between school climate and social capital	Positively correlated	N/A		Cronbach’s alpha	Subscales range 0.79–0.88	N/A	
Geraee et al., (2019) [33]	EFA and CFA	Four-factor solution, factor loadings > 0.5. RMSEA 0.04, CFI 0.87	Inverse mediating role of social media use in relationship between social capital and life satisfaction	*p* < 0.001	N/A		Cronbach’s alpha	Subscales range 0.69–0.94	N/A	
Hall et al., (2014) [34]	N/A		Protective effect on depressive symptoms and functional impairment	*p* < 0.001	N/A		Cronbach’s alpha	Total scale 0.70	N/A	
Harpham et al., (2005) [24]	Factor analysis	Eight-factor solution, no specifics	Inverse relationship with common mental disorders	Not confirmed	N/A		N/A		Test–retest, Spearman’s correlation coefficient for reliability	67% of SC-items had >0.70
Khawaja et al., (2006) [35]	N/A		Positive relationship with self-rated health	Confirmed	N/A		Cronbach’s alpha	Total scale 0.59	N/A	
Krasny et al., (2013) [36]	N/A		Hypothesis that EE programs would increase social capital	Partly confirmed	N/A		Cronbach’s alpha, KR	Social trust 0.64, informal socializing 0.74, Diversity of friendship KR-20 0.71, no test for other subscales	Test–retest	No mean score difference in control group
Lau & Li (2011) [37]	N/A		Positive relationship with well-being	*p* < 0.001	N/A		Cronbach’s alpha	Family social capital 0.84, School social capital 0.70	N/A	
Magson et al., (2014) [38]	CFA	Final model: six factor solution, loadings 0.46–0.81, RMSEA 0.042, CFI 0.98.	Inverse relationship with mental health outcomes	*p* < 0.001	Invariance testing CFA models for gender and regions.	Gender: CFI change <0.1, RMSEA 0.069, Region: CFI change > 0.1	Cronbach’s alpha	Subscales range 0.70–0.89	N/A	
Onyx et al., (2005) [39]	Hierarchical factor analysis with varimax rotation	One secondary- and seven primary-factor solution, loadings range 1.56–3.86, eigenvalues 1.32–5.81, Total variance explained 48.6%, RMSEA 0.08	Factors compared with adult sample who completed scale by Onyx & Bullen (2000)	Both recurring and unique factors, not further specified	Invariance testing for age groups	Three factors displayed significant differences between age groups, specifics not reported	Cronbach’s alpha	Total scale 0.83	N/A	
Paiva et al., (2014) [40]	EFA with Varimax rotation and CFA	Four-factor solution, loadings > 0.48, eigenvalues 1.15–2.89, Variance explained 61.68%. KMO 0.63, CFI: 0.85, RMSEA 0.105	N/A		N/A		Cronbach’s alpha	Total scale 0.71	Test–retest with Kappa-coefficient	Range 0.64–0.97
Pourramazani et al., (2019) [41]	N/A		Inverse relationship with substance use	Partly confirmed	N/A		Cronbach’s alpha	Subscales range 0.62–0.79	N/A	
Ryan & Junker (2019) [42]	CFA	Model A: Loadings 0.39–0.90, CFI 0.93, RMSEA 0.06 Model B & C: Loadings 0.30–0.90, CFI 0.94, RMSEA 0.06	Hypothesis of consistency with theory	Confirmed through CFA with caution for sample size	Invariance testing for lunch prize subsidies and age	*p* ≤ 0.05	Cronbach’s alpha	Subscales range 0.72–0.80	N/A	
Takakura et al., (2014) [43]	EFA with promax rotation for cognitive subscale	Two-factor solution, eigenvalues 2.4 & 5.7, item loadings 0.48–0.94, variance explained 68.1%	Positive relationship with self-rated health and physical activities, inverse relationship with depressive symptoms	Cognitive: 0.15–0.31/−0.25–−0.39, *p* < 0.001. Structural not confirmed	Tested for correlation with safety	Cognitive: 0.26–0.63, *p* < 0.01. Structural not confirmed	Cronbach’s alpha	School 0.92, Neighborhood 0.94	Test–retest with Pearson correlation coefficient	Range 0.48–0.81
Wang & Gu (2019) [44]	N/A		Positive relationship between online SC and peer relationships and academic identity	*p* < 0.01	N/A		Cronbach’s alpha	Total scale 0.84, subscales 0.77 and 0.78	N/A	

N/A = Not Applicable. EFA = Exploratory Factor Analysis, CFA = Confirmatory Factor Analysis, PCA = Principal Component Analysis, VFA = Verifying Factor Analysis, ICC = Intra-Correlation Coefficient. CFI = Comparative Fit Index, RMSEA = Root Mean Square Error of Approximation.

## Data Availability

Not applicable.

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
