# Peer review of "Current Conceptualization and Operationalization of Adolescents’ Social Capital: A Systematic Review of Self-Reported Instruments"

_ijerph, 2022, doi:10.3390/ijerph192315596_

Round 1

Reviewer 1 Report

Thank you very much for this valuable manuscript. I really appreciate the idea of the paper and the way you presented the results of your review. All in all, I strongly support the publication of the manuscript. However, before doing so, I recommend a minor revision of the following points:

Table 1, column "Study design": The categories are not used uniformly, as a study can be both cross-sectional and exploratory, or simultaneously follow a mixed methods approach.

Table 4, "Face and content validity..." column. What is meant by theory explicit ? It is also not described in section 3.4.1 or somewhere else in the text. This column also has the weakness of subsuming different categories and information.

Discussion: I appreciate the reference to the heterogeneity and multidimensionality of the construct social capital, which can hardly be captured with only one scale or instrument. This information also prevents research from creating further déjà variables. However, I would like to read some sort of recommendation on which aspects of the construct the authors would like to see at the core of social capital. If this is not possible due to space and work constraints, an indication that this is exactly what is still missing in the research would be very nice. 

Reviewer 2 Report

Thank you for the chance to review this manuscript on the conceptualisation of adolescent social capital. This is closely related to my research interests and is, on the whole, a well-written and interesting manuscript. 

I do however, of course, have a few comments that I hope will improve the overall manuscript and the analysis within it. I share feedback for the individual sections below.

Introduction

The introduction is well written and provides a clear case for your study. Overall, I think the scope of the introduction and associated debate would be improved if you established a clear definition of social capital and, later, illustrated some of the 'multiple dimensions or constructs' with examples. 

Results

In the dimensions, you mention that "social cohesion" was a construct measured in studies. Yet social cohesion is arguably an even larger concept than social capital, as it typically takes into account social capital (i.e. social relations), but also items like trust in institutions, perceptions of fairness, mutual tolerance, and more. 

I realise you may simply be reporting on the constructs as they are presented in the articles themselves, but I suggest that it would be valuable to disentangle this in the results or discussion. I provide a few useful general references on social cohesion below. Full disclosure, one of them is from myself, but I in now way expect or require it to be used - it is just to illustrate the conceptual confusion.  

Likewise, I notice in Table 2 that some constructs seem to lean to a very individual view (cognitive social capital, self-concept). Yet you also present social capital as something interpersonal. Is there a certain 'displacement of scope' within some instruments, whereby they measure individual skills and belief instead of social capital? 

More generally, I suppose I would be interested to know how conceptually coherent the instruments were. 

Discussion

You talk about the different contexts of measurement, but I am not sure the nuances come through clearly. How does one differentiate between 'peers' and 'school' for example? Can I not have peers at school? Maybe I missed this elsewhere in the text, but I'd like to know how these differences are defined. 

Your points regarding the need to involve adolescents make a lot of sense. The top-down presentation/definition of social capital can obscure local meaning or nuances, especially for younger groups. For instance, such conceptualisations may ignore the increasingly informal or digital ways youth engage civically (see, e.g, Harris, 2010). 

As I highlighted above, I think it could be worthwhile to also discuss some potential conceptual confusion between social capital and other constructs like social cohesion or certain individual-level measures. 

Overall, congratulations on an interesting review. As promised, I add a few potentially helpful further references below.

---

Dragolov, G., Ignácz, Z. S., Lorenz, J., Delhey, J., Boehnke, K., & Unzicker, K. (2016). Theoretical Framework of the Social Cohesion Radar. In G. Dragolov, Z. S. Ignácz, J. Lorenz, J. Delhey, K. Boehnke, & K. Unzicker (Eds.), SpringerBriefs in Well-Being and Quality of Life Research. Social Cohesion in the Western World (pp. 1–13). Cham: Springer International Publishing. https://doi.org/10.1007/978-3-319-32464-7_1

Harris, A., 2010. Young People, Everyday Civic Life and the Limits of Social Cohesion. Journal of Intercultural Studies, 31 (5), 573–589.

Moustakas, L. (2022). A Bibliometric Analysis of Research on Social Cohesion from 1994–2020. Publications, 10(1), 5. https://doi.org/10.3390/publications10010005

Schiefer, D., & van der Noll, J. (2017). The Essentials of Social Cohesion: A Literature Review. Social Indicators Research, 132(2), 579–603. https://doi.org/10.1007/s11205-016-1314-5
